# Novel Cold-Adapted Recombinant Laccase KbLcc1 from *Kabatiella bupleuri* G3 IBMiP as a Green Catalyst in Biotransformation

**DOI:** 10.3390/ijms22179593

**Published:** 2021-09-04

**Authors:** Katarzyna M. Wiśniewska, Aleksandra Twarda-Clapa, Aneta M. Białkowska

**Affiliations:** Institute of Molecular and Industrial Biotechnology, Lodz University of Technology, Stefanowskiego 2/22, 90-537 Łódź, Poland; kma.wisniewska@gmail.com (K.M.W.); aleksandra.twarda-clapa@p.lodz.pl (A.T.-C.)

**Keywords:** psychrophilic laccase, *Kabatiella bupleuri*, recombinant enzyme, *Pichia pastoris*, decolorization, biotransformation, vanillin

## Abstract

Cold-adapted enzymes are useful tools in the organic syntheses conducted in mixed aqueous-organic or non-aqueous solvents due to their molecular flexibility that stabilizes the proteins in low water activity environments. A novel psychrophilic laccase gene from *Kabatiella bupleuri*, G3 IBMiP, was spliced by Overlap-Extension PCR (OE-PCR) and expressed in *Pichia pastoris*. Purified recombinant KbLcc1 laccase has an optimal temperature of 30 °C and pH of 3.5, 5.5, 6.0, and 7.0 in the reaction with 2,2′-azino-bis(3-ethylbenzothiazoline-6-sulfonic acid) (ABTS), guaiacol, sinapic acid, and syringaldazine, respectively. Moreover, laccase KbLcc1 is highly thermolabile, as it loses 40% of activity after 30 min at 40 °C and is inactivated at 50 °C after the same period of incubation. The new enzyme remained active with 1 mM of Ni^2+^, Cu^2+^, Mn^2+^, and Zn^2+^ and with 2 mM of Co^2+^, Ca^2+^, and Mg^2+^, but Fe^2+^ greatly inhibited the laccase activity. Moreover, 1% ethanol had no impact on KbLcc1, although acetone and ethyl acetate decreased the laccase activity. The presence of hexane (40%, *v*/*v*) caused a 58% increase in activity. Laccase KbLcc1 could be applied in the decolorization of synthetic dyes and in the biotransformation of ferulic acid to vanillin. After 5 days of reaction at 20 °C, pH 3.5, with 1 mM ABTS as a mediator, the vanillin concentration was 21.9 mg/L and the molar yield of transformation reached 14.39%.

## 1. Introduction

Laccases (EC 1.10.3.2, benzenediol:oxygen oxidoreductases) are multicopper oxidases which catalyze the oxidation of a broad range of substrates, e.g., phenolic and polyphenolic compounds, polyamines, aryl diamines, aminophenols, and lignins. Depending on the value of redox potential (RP), two groups of laccases are distinguished, namely low RP (0.4–0.6 V) and high RP (0.6–0.8 V) [1]. When the redox potential of laccase is lower than the RP of the substrate, the enzyme cannot oxidize it. In that situation, it is necessary to use small molecules capable to act as electron transfer mediators, such as 2,2′-azino-bis (3-ethylbenzothiazoline-6-sulfonic acid) (ABTS) or 1-hydroxybanzotriazole (HBT). Mediators added to reaction mixtures enhance the use of these enzymes for industrial processes, including the detoxification of industrial dyes, delignification of lignocellulosic biomass, and bioremediation of xenobiotic compounds, polycyclic aromatic hydrocarbons, and pesticides [2].

The application of a laccase in industrial and environmental processes requires a large amount of an enzyme. Therefore, it is important to identify a highly efficient microorganism or to develop an efficient expression system to produce a heterologous laccase. For the expression of recombinant laccases and laccase-like enzymes, several expression systems based on bacteria (*Escherichia coli*), plants (*Arabidopsis thaliana*), yeast (*Pichia pastoris*, *Saccharomyces cerevisiae*), and filamentous fungi (*Aspergillus niger*, *Aspergillus oryzae*, *Trichoderma resei*) have been used. Considering that these enzymes are glycoproteins that require glycosylation for their activity, bacterial systems are used less often [3]. Due to the fact that the optimization of gene expression by directed evolution methods is complicated to achieve in plants and filamentous fungi, the cloning and expression of fungal laccases and laccase-like enzymes is usually performed in the yeast [4]. The main advantages of the yeast systems include growth on solid medium as individual colonies, absence of laccase genes to be expressed, and the secretion of recombinant proteins into the medium, providing simple activity detection without a lysis step. The most commonly used yeast species for laccase and laccase-like enzyme production are *S. cerevisiae* and *P. pastoris* [5].

Laccases show great potential for use in many industrial processes, including the paper industry for bleaching pulp [6], food industry for wine stabilization by removing excessive phenolic compounds [7], in the processes of detoxification [8,9] and decolorization [10,11] of sewage sludge, as well as in the degradation of xenobiotics [12,13,14]. The use of laccases in organic syntheses is particularly interesting and promising. Laccase-mediated processes are used, inter alia, for the preparation of geranial, decanal, and 3,4-dimethoxybenzyl aldehyde [15]. In turn, the reactions of direct oxidation by laccase are used for the dimerization of resveratrol [16], rutin, and epigallocatechin gallate [17,18], as well as for the production of coloring substances for the cosmetics [19], textile [20], and food industries [21].

Vanillin (4-hydroxy-3-methoxybenzaldehyde) is the main flavor of vanilla pod (2.5–4.5% of dry weight). To date, the most important source of vanillin remains chemical synthesis using guaiacol and lignin as a starting material. Recent advances in biotechnology have allowed for a bioengineering approach to the synthesis of vanillin. These new methods are based on the bioconversion of certain natural substrates, such as lignin, ferulic acid, eugenol, isoeugenol, and phenolic stilbenes, etc., using microorganisms such as fungi, bacteria, plant cells, or their genetically engineered counterparts [22]. The bioconversion of ferulic acid is one of the most intensively studied process of vanillin production. Many microorganisms, including *Streptomyces sannanensis* [23], *A. niger* [24], *Bacillus subtilis* [25] *Amycolatopsis* sp. [26], and *Pycnoporus cinnabarinus* [27], have been described as potential producers of vanillin from ferulic acid. Moreover, Furuya and coworkers studied the production of vanillin by a coenzyme-independent decarboxylase/oxygenase two-stage process [28].

In this study, a new cold-adapted laccase-like enzyme from *Kabatiella bupleuri* G3 IBMiP was expressed in *P. pastoris*. In our previous work, a novel laccase-producing yeast-like fungus, *K. bupleuri* G3 IBMiP, was identified, and an efficient secretion of the new enzyme was optimized [29]. Now, the coding sequence of KbLcc1 gene was obtained using Overlap-Extension PCR (OE-PCR), known as splicing in vitro. The recombinant protein was purified and characterized in terms of the optimal reaction conditions (pH and temperature), stability at different pH and temperature, in the presence of metal ions and organic solvents, and substrate specificity. The last part of study describes the application of KbLcc1 laccase-like enzyme in the decolorization of synthetic dyes and the transformation of ferulic acid into vanillin.

## 2. Results and Discussion

### 2.1. Expression of KbLcc1 Laccase from K. bupleuri G3 IBMiP

The genome of the psychrophilic yeast-like fungal strain (*K. bupleuri* G3 IBMiP) was sequenced and assembled in the SPAdes programme into 207 contigs of total 23,395,295 bp. N50 value was equal to 193,564 bp. Further, 10,172 gene-coding sequences were predicted in the Augustus programme, and a functional annotation was found for most of them.

The amino acid sequences were searched for the presence of domains characteristic for the investigated laccase in the Pfam database and Conserved Domain Database (CDD). Nine proteins containing the copper-binding domains were found, but only four of them contained domains characteristic for laccases (i.e., cd13854, cd13880, and cd13901 from CDD). Only three of them contained the signal peptide necessary for secretion of the enzyme. The extracellular psychrophilic laccase produced by *K. bupleuri* G3 IBMP was initially characterized to have a molecular mass of about 72 kDa [29]. Therefore, the predicted masses of three potential laccases found in the genome were compared. The nucleotide sequence of KbLcc1 was found to contain an open reading frame (ORF) of 1938 bp capable of coding for a protein of 649 amino acid residues, which corresponded to the molar weight of about 69 kDa. Domains characteristic for the laccases and copper binding sites were both found in this sequence. A comparison with the amino acid sequences database (non-redundant in GenBank) presented 86% similarity of the enzyme from *K. bupleuri* to laccase-like multicopper oxidase from *Aureobasidium subglaciale* (XP_013344102.1) with 100% query coverage, whereas a comparison with the SWISSprot database (containing the sequences verified by the experts) revealed 59% similarity to the laccase from *Neurospora crassa* (P06811.3) with 83% query coverage. Finally, the sequence of the reported enzyme KbLcc1 was aligned with eight sequences of laccases produced by fungi *Ascomycota*, including the enzyme from *A. subglaciale* that displayed the highest similarity to KbLcc1. Multiple sequence alignment allowed for the identification of four conservative regions in the KbLcc1 sequence that are characteristic for laccases and responsible for binding of the copper ions at the active site (Table 1, Figure 1).

Gene coding for the protein of interest contained one intron which had to be removed. To this end, the OE PCR method, also described as in vitro splicing, was applied (Figure 2). This is a simple, versatile technique for site-directed mutagenesis and gene splicing. Initial PCRs generate overlapping gene fragments that are then used as a template DNA for the next PCR to create a full-length product. Internal primers generate overlapping, complementary 3′ ends on the intermediate fragments and encode the nucleotides found at the junction of adjoining gene fragments. Overlapping strands hybridize at these 3′ end fragments in a subsequent PCR and are extended to the full-length coding sequence amplified by flanking primers. These primers can include restriction enzyme sites for further inserting the PCR product into an expression vector [30].

The KbLcc1 coding sequence flanked on 3′ and 5′ with the restriction sites for StuI and FseI restrictases was cloned into the yeast expression vector pPINKα-HF containing the α-factor sequence, thereby enabling efficient protein secretion (construct scheme presented on Figure 3a).

Restriction analysis confirmed the correctness of the cloning–digestion with SpeI linearized the vector into a DNA molecule of 9827 bp, whereas the multiple digestions with SpeI, AscI, NdeI and EcoRI lead to the creation of five fragments of lengths: 1131, 1412, 1836, 2321, and 3127 bp (Figure 3b). Due to the wide length range of the observed DNA fragments, it was impossible to accurately determine the length of the linear vector molecule with the insert. Nevertheless, the electrophoretic image shows that all vector particles have been digested, only one band is visible and no bands from circular particles are visible. In the case of multi-enzyme digestion, the estimated lengths of the fragments visible in the electrophoretic image correspond to those expected. The correctness of the insert sequence was further confirmed by sequencing and comparison with the sequence from the genome, which gave 100% identity.

The recombinant strains *P. pastoris* harboring pPINKα-HF-KbLcc1 were cultured in BMMY medium with methanol added daily to a final concentration of 0.5% to maintain enzyme induction. Laccase activity in the supernatant of the post-culture liquid reached 13.6 U/L after 120 h for clone CI. SDS-PAGE analysis showed that the secreted expression took place after 48 h, and a constant rise in the amount of the recombinant protein was observed until 120 h (Figure 4a).

The best recombinant clone CI was chosen for the optimization of culture conditions. Various copper (II) ion concentrations (0.5, 1.0, 1.5, 2.0, and 2.5 mM) were added to buffered complex methanol medium (BMMY) medium. Increase of the laccase activity was observed for 0.5–2.0 mM, whereas 2.5 mM turned out to be too high (the laccase activity dropped to about 30 U/L after 7 days; Appendix A). The highest enzymatic activity was noted after 7 days of the culture with 2.0 mM of Cu^2+^, namely 127.7 U/L, which was about 10 times higher than in medium without supplementation after 5 days (13.6 U/L) (Figure 4b). The reasons for the increased activity can be found in the fact that copper ions added to the substrate during heterologous laccase production ensure the correct folding of the enzyme [31]. The optimization of the concentration of copper ions in the medium also allowed for an increase in the efficiency of expression in the case of laccase Cplcc1 from *Coriolopsis polyzona* MUCL 38443. The addition of 0.8 mM CuSO_4_ increased the activity from 211.29 U/L to 380.52 U/L. Moreover, further optimization, including the concentration of methanol and alanine in the medium and the temperature of the culture, increased the activity of heterologous laccase to a maximum value of 800 U/L [32]. In turn, Bao et al. [33] achieved the expression of laccase Lac1 from *Coprinus comatus* in *P. pastoris* at the level of activity of 550 U/L in a medium containing 0.5 mM Cu^2+^ and 1% methanol [33]. The KbLcc1 laccase expression efficiency at the level of approximately 130 U/L is sufficient for the characterization of the enzyme and the search for areas of its application. However, the production yield of KbLcc1 could be further improved by optimization of the expression medium, culture conditions, and genetic engineering strategies, such as the usage of different promoters, secretory signal sequences, and codon optimization for the *P. pastoris* expression system.

### 2.2. Purification of Recombinant Laccase

Post-culture liquid after 168 h of culture of *P. pastoris* clone CI was centrifuged, concentrated, and purified to obtain a highly pure preparation of laccase KbLcc1. Thus, 48.94-fold purification was achieved, which correlated to a final enzyme yield of 29.39% (Table 2). The protein concentration after purification was 0.359 mg/mL. Both the crude and purified enzyme were stored in a refrigerator at 4 °C. Expression of the recombinant psychrophilic laccase KbLcc1 resulted in 40× higher enzymatic activity, from the level of 0.5 U/mg for native [29], to the level of 20.5 U/mg for the heterologously expressed protein. Similar values were achieved for the recombinant laccase LccA from *Trametes versicolor* (28.8 U/mg) [34]. In turn, the cold-adapted laccase from *Colletotrichum lagenarium* expressed in *P. pastoris* was characterized with lower specific activity of 5.33 U/mg [35].

The SDS-PAGE electrophoresis of the purified recombinant enzyme corroborated the high efficiency of the purification process, showing the expected protein band without clearly visible other ballast protein bands (Figure 4c). The recombinant enzyme had a molecular weight of 76 kDa which is slightly higher than the theoretical mass of 69.6 kDa probably due to glycosylation. Analysis of the KbLcc1 laccase coding sequence revealed the presence of four potential N-glycosylation sites. The molecular mass of the purified laccase KbLcc1 (76 kDa) is typical for fungal laccases which may fall in the range of 50–140 kDa, although the most common enzymes range from 60 to 70 kDa [36]. The content of sugar residues in laccase molecules most often varies between 10% and 25% and depends on the type of microorganism and the growth environment. For example, laccase of marine fungi *Trematosphaeria mangrovei* isolated from decayed wood samples collected from Abour Keer in Egypt is a glycoprotein with 23% carbohydrate content and molecular weight of 48 kDa [37]. The presence of sugar residues protects the enzyme against proteolysis and inactivation by radicals formed as a result of laccase action. Moreover, it also ensures laccase conformational stability [38]. Molecular masses of laccases from *Aureobasidium pullulans* strains NRRL-50381 and NRRLY-2568 were greater than 100 and 150 kDa, respectively. After endoglycosidase treatment, molecular weights were reduced to 60–70 kDa for NRRL-50381 and to over 100 kDa for NRRLY-2568, but both enzymes lost 40–50% of original activity [39].

### 2.3. Characterization of Purified KbLcc1 Laccase

#### 2.3.1. Influence of pH and Temperature on Purified Laccase KbLcc1 Activity and Stability

Optimal pH and temperature values were determined for the reactions with four substrates: ABTS, syringaldazine, guaiacol, and sinapic acid. Optimal pH was determined to be 3.5, 7.0, 5.5, and 6.5, for ABTS, syringaldazine, guaiacol, and sinapic acid, respectively (Figure 5a). The impact of pH on the stability of the laccase was investigated using ABTS as a substrate. The enzyme is stable for 24 h over a wide pH range, retaining more than 90% activity at pH 4.5–6.0, and more than 80% at pH 3.5–8.5 (Figure 5b). These values are similar to those observed for the native laccase G3 *K. bupleuri* [29], as well as to other fungal laccases. ABTS is the most popular substrate used in the study of laccase activity. The optimum pH values for such reactions are most often in the range of 3.0–4.0 [35,40], however, there are also laccases characterized by slightly higher optimal pH reaching the values of 5.0, e.g., for the laccase from *A. pullulans* [41] and *Thielavia* sp. [42], and pH 6.0 for the laccase from *Thermus thermophilus* [43]. Optimal pH in the reaction with syringaldazine, sinapic acid and guaiacol are most often higher than for ABTS, but they remain within the acidic pH range. The process of oxidation of phenolic compounds depends on the difference in the redox potential between the reactant and the T1 copper atom in the enzyme’s active site. As the pH increases due to the oxidative proton release, the E_0_ potential of the phenolic compounds decreases. The change in E_0_ in the case of phenol can even reach 490 mV with an increase in pH from 2.7 to pH 11. At the same time, for laccases, it is much smaller (approximately 100 mV). Moreover, an increase in pH causes OH^−^ to bind to copper atoms T2/T3, which disrupts the electron transfer between T1 and T2/T3 centers [44].

The highest activity of laccase KbLcc1 towards syringaldazine was noted at pH 7.0, which is similar to the laccase from *Melanocarpus albomyces* [45] and higher than for the laccase from *Vovariella volvacea* V14 (optimal pH 5.6) [46] or from *Shiraia* sp. SUPER-H168 (optimal pH 6.0) [47].

The highest activity with ABTS as a substrate was recorded at 30 °C (Figure 5c). A rapid drop in laccase activity was observed above 45 °C. The enzyme was operating at low temperatures, 0–20 °C, retaining 34–87% of the maximum activity. The same optimal temperature, i.e., 30 °C, was observed for all four substrates (Table 3, Appendix A), which was consistent with that reported for the native G3 laccase from *K. bupleuri*. Recombinant laccase KbLcc1 is stable at temperatures of 4–30 °C for 120 min (Figure 5d). Higher temperature probably causes irreversible changes in the protein structure, gradually leading to its inactivation. The decrease in activity was observed after 30 min at 40 °C by almost 40%, while after 120 min the enzyme retained only 20% of its initial activity. Moreover, the temperature of 50 °C almost completely inactivated the enzyme after 30 min of incubation (Figure 5d). The results obtained for the recombinant KbLcc1 laccase, i.e., high activity at low temperatures (61% and 34% of maximal activity at 10 °C and 0 °C, respectively) and high thermolability, confirm the psychrophilic nature of the new enzyme.

Most of the described laccases are stable at temperatures of 30–50 °C, and they lose their activity above 60 °C. Despite the relevance of cold-adapted and psychrophilic laccases for industrial and environmental applications, studies related to these proteins are scarce. Laccase produced by *Cryptococcus albidus* (now *Naganishia albida*) was characterized by maximum activity in the temperature range of 20–30 °C and high thermolability. Moreover, its half-life was 81 min at 25 °C and 64 min at 45 °C [48]. Laccase from *T. mangrovei* is an interesting example as well, because despite maximum activity at 65 °C, it loses 40% of activity already after 60 min at 40 °C, and 98% at 65 °C [37]. Laccase Lac1326 derived from a marine metagenomic library can also be categorized as a cold-adapted enzyme. Maximum activity was noted for 60 °C. However, it retained 26.5% activity at 0 °C and more than 40% at 10 °C [49]. Cold-adapted enzymes are used in organic syntheses carried out in mixed aqueous-organic or non-aqueous solvents. The inherent flexibility of psychrozymes counteracts the stabilizing effects of low water activity in organic solvents [50].

#### 2.3.2. Substrate Specificity

Most laccases are characterized by a strong affinity and a high catalytic constant for ABTS and syringaldazine, while guaiacol oxidation is slower and the K_M_ value is higher. Additionally, K_M_ values are usually low for sinapic acid [44]. The reported recombinant laccase followed this trend (Table 3).

K_M_ values for ABTS are usually between 0.1 and 3.7 mM [44]. For laccases from *Ganoderma lucidum* expressed in *P. pastoris*, K_M_ and V_max_ values were 0.521 mM and 19.65 U/mg, respectively, [51] which is very similar to the enzyme from this study. Laccase KbLcc1 displayed a strong affinity to syringaldazine (K_M_ = 0.021 mM, k_cat_ = 38.31 s^−1^), with subsequent low affinity to guaiacol (K_M_ = 13.28 mM, k_cat_ = 5.45 s^−1^). The highest V_max_ was noted for ABTS, i.e., 34.18 U/mg (Table 3). Similar affinity to syringaldazine was reported for the enzyme from *Th. thermophilus* (K_M_ = 0.017 mM, k_cat_ = 10.3 s^−1^) [43]. An et al. [52] described the laccase produced by *Postia placenta* MAD-698-R, which was not active towards syringaldazine, but rather towards ABTS (K_M_ = 0.022 mM, k_cat_ = 1213 s^−1^) [52]. The broad substrate specificity indicates a high potential for the use of KbLcc1 laccase in organic synthesis, dye decolorization, and bioremediation.

#### 2.3.3. Influence of Ions on Laccase KbLcc1 Activity

The influence of metal ions on enzyme activity is extremely important in the context of industrial applications. As a result of their presence, catalytic proteins can be both activated and inhibited, which in turn determines the positive course of the process or the absence of the enzyme’s action.

The influence of various cations on the activity of purified laccase in the ABTS oxidation reaction as well as the stability of the enzyme during 24-h and 10-day incubations in the presence of selected ions was investigated. Incubation was carried out at 4 °C. Thereby, Ni^2+^, Cu^2+^, Mn^2+^, and Zn^2+^ ions at the concentration of 1 mM, as well as Co^2+^, Ca^2+^, and Mg^2+^ ions at 2 mM did not have any significant impact on the activity of laccase KbLcc1. Moreover, for these ions at 5 mM concentration, the activity of at least approximately 80% of the maximum was still observed. For higher concentrations, a gradual decrease in activity was observed, up to approximately 60% in the presence of 10 mM Co^2+^. Only iron ions were found to have a strong negative effect on enzyme activity as 1 mM Fe^2+^ and 2 mM Fe^3+^ completely inhibited enzyme’s action (Figure 6).

After 24-h incubation, a clear decrease in laccase activity was observed in the presence of cations Fe^2+^ and Fe^3+^ (Figure 7). Indeed, a 1 mM concentration resulted in a loss of 70.1% and 89.3% of activity, respectively. Higher concentrations almost completely inactivated the enzyme. Ni^2+^ ions at 1 mM caused a minor 8.2% enzyme’s activation, whereas Cu^2+^ ions lead to approximately 7% drop at the concentrations of 1 and 2 mM, similarly to 5 mM Co^2+^. In the case of Mn^2+^ and Zn^2+^ ions, a significant negative effect was observed only for 2 mM, whereas Ca^2+^ and Mg^2+^ ions did not have any impact on the stability of KbLcc1 laccase. Larger differences were found after 10-days incubation (Appendix A). In the samples containing 5 mM Ca^2+^ and 2 and 5 mM Mg^2+^, a significant increase in activity was observed, by 8.8%, 11.2%, and 10.1%, respectively. It was found that these ions have a beneficial effect on the stability of the tested laccase, which can be used during several days of enzymatic reactions, as well as for longer storage. The prolonged incubation confirmed a slight negative effect of the presence of manganese ions (a decrease in activity by approximately 9% at concentrations of 1 and 2 mM) and a more pronounced impact in the case of copper and zinc ions, which reduced the activity by approximately 20% in the tested concentrations. The influence of nickel and cobalt ions also turned out to be negative at 1, 2, and 5 mM, as nickel ions decreased laccase activity by 8.7%, 20.1%, and 27.9%, respectively, and cobalt ions by 4.3%, 16.6%, and 37.4%, respectively.

The presence of metal ions influences the activity of laccases, both by inducing and reducing it. Metal ions such as copper, calcium, nickel, cobalt, and manganese, as a rule, have a noticeably inducing effect on the enzyme activity [53]. In the case of KbLcc1 laccase, such an effect was observed for Ni^2+^ ions at the concentration of 1 mM during the 24-h incubation and for all tested Ca^2+^ concentrations, when the incubation time was extended to 10 days. The enzyme produced by *Schizophyllum commune* NI-07 retained the activity in the presence of 0.1 mM Cu^2+^ (after 10 min incubation with metal ions followed by activity assay), and the other tested ions had a negative impact for the laccase even at 0.01 mM [54]. Laccase from *T. mangrovei* retained approximately 60% activity in the presence of Co^2+^ and Mg^2+^ ions and approximately 70% in the presence of K^+^ and Na^+^ ions during 15 min incubation, whereas ions Ca^2+^ and Mn^2+^ had very little impact on the activity. In turn, Fe^2+^ ions at 1 mM completely inactivated the enzyme [37]. Laccase produced by fungi *Scytalidium thermophilum* retained its activity in the presence of 10 mM Cu^2+^, Mg^2+^, Zn^2+^, Ni^2+^, Co^2+^, Ca^2+^, and Mn^2+^ ions, whereas Fe^2+^ ions inhibited the enzyme by 98% [55]. The negative effect of iron ions, especially Fe^2+^, on the activity of both KbLcc1 laccase and others, is probably due to the fact that they can disrupt the electron transport system in laccase, resulting in inhibition of the enzyme [56].

#### 2.3.4. Influence of Organic Solvents on Laccase KbLcc1 Activity

Multicopper oxidases are used in the reactions of organic syntheses. For this reason, the influence of the presence of organic solvents in the reaction environment on the activity of the new psychrophilic laccase KbLcc1 was investigated. Acetone, ethyl acetate, methanol, ethanol, isopropanol, and hexane at concentrations of 1%, 10%, 20%, and 40% (*v*/*v*) were selected for the experiment (Figure 8). In addition, the stability was analyzed by 24-h incubation of the enzyme with the tested solvents at 4 °C, followed by the ABTS oxidation reaction under the optimal conditions (Figure 9).

The presence of 1% (*v*/*v*) ethanol in the reaction mixture did not affect the performance of the purified KbLcc1 laccase. A slight negative effect was observed for 1% (*v*/*v*) methanol and 1% (*v*/*v*) isopropanol (approximately 90% of maximum activity), and a greater effect for 1% (*v*/*v*) acetone and 1% (*v*/*v*) ethyl acetate (approximately 70% of maximum activity). A further increase of the acetone concentration resulted in over 80% inactivation of the enzyme. For methanol, ethanol, and isopropanol, a similar effect was observed only at 40% (*v*/*v*) concentration, while in the presence of 40% (*v*/*v*) ethyl acetate, laccase maintained approximately 20% of its maximum activity. The addition of hexane to the reaction mixture had a positive effect on the oxidation of ABTS by laccase KbLcc1. Hence, 1% (*v*/*v*) hexane caused a 16% increase in activity, while the greatest positive effect was observed for 40% (*v*/*v*) hexane (an increase in activity by 58%).

In the case of testing the enzyme stability in organic solvents, it was observed that pre-incubation in the presence of 1% (*v*/*v*) of the tested solvents had little effect on the stability of laccase (over 80% of residual activity), but a concentration increase in most cases had a strong negative effect. Ethyl acetate caused a loss of activity by 65% already at 10% (*v*/*v*) of the additive, and in the case of 20% (*v*/*v*) it completely inactivated the enzyme. The effect of pre-incubation with methanol was not so unfavorable, as at 40% (*v*/*v*) concentration the laccase retained 72% activity in relation to the control sample. The negative effect of acetone and ethanol was also observed for enzymes from *Pycnoporus sanguineus* CS43. For the two laccases LacI and LacII, there was a loss of 50% activity in the presence of 47% and 31% (*v*/*v*) acetone and 55% and 52% (*v*/*v*) ethanol, respectively [57]. The presence of ethanol, methanol, and hexane in a concentration of 20% (*v*/*v*) did not affect the activity of archaean laccase from *Halopiger aswanensis* strain ABC_IITR. Increasing the hexane concentration to 40% (*v*/*v*) resulted in a rise in activity by 23%. Acetone had a positive effect on the enzyme, causing a 28% and 45% increase in activity at a concentration of 20% and 40% (*v*/*v*), respectively [58]. On the other hand, for TpL laccase from *Trametes polyzonia* WRF03, a slight negative effect of polar solvents at a concentration of 10% (*v*/*v*) was found, similarly to KbLcc1 laccase, while 50% (*v*/*v*) solvent addition resulted in a loss of almost 85% of activity in the case of ethanol, 55% for methanol, and 50% for acetone after 1 h of incubation [59]. Yang et al. [49] obtained opposite results for the laccase-like enzyme Lac1326 from a marine metagenomic library after 12-h incubation. Polar solvents, such as methanol, ethanol, and acetone at a concentration of 10% (*v*/*v*) favorably influenced the enzyme. There was a 60%, 70%, and 137% increase in laccase activity, respectively [49]. Among the tested solvents, only hexane positively influenced the activity of the KbLcc1 enzyme, i.e., the 20% (*v*/*v*) concentration of the solvent resulted in a 30% increase in activity, while at 40% (*v*/*v*) hexane, a positive 7% effect was still observed. Depending on the nature of the organic solvent, its effect on the stability of laccase’s catalytic hydrophobic core residues and ability to retain hydration properties may be different. Jafari et al. [60] investigated the effect of the presence of organic solvents on the hydration shell, structure, dynamics, interaction, and activity of laccases on the example of the laccase from *T. versicolor*. The enzyme in the presence of 50% (*v*/*v*) hexane retained a stable conformation (tertiary and secondary structures). On the other hand, 50% (*v*/*v*) ethanol caused changes in beta-turn and beta-sheet elements, negatively affecting the structure of laccase. Moreover, in the presence of hexane, the availability of substrate molecules to the active site increased, which increased the values of V_max_ and k_cat_ in comparison to the reaction conducted in water or 50% (*v*/*v*) ethanol (decreased V_max_ and k_cat_ values). One possible explanation comes from the fact that polar solvents with logP value smaller than 2 (i.e., ethanol) may remove the essential water molecules from the hydration shell of the enzyme, which is crucial for retaining its activity [60].

### 2.4. Application of the Recombinant Laccase from K. bupleuri

#### 2.4.1. Decolorization of Synthetic Dyes

Our previous research has shown the possibility of effective use of the native G3 laccase from *K. bupleuri* in the decolorization of dyes such as methylene blue, alkaline fuchsin, Coomasie Brilliant Blue R-250 and crystal violet. The highest efficiency was found for crystal violet, i.e., about 40% for 1 h at 30 °C [29]. 

Heterologous expression of the gene encoding the KbLcc1 laccase and purification of the protein allowed to obtain an enzyme with a higher specific activity than the native one, and, at the same time, remarkably similar in terms of optimal reaction conditions (pH, temperature) and stability (pH-stability, thermostability). The process of decolorization of methylene blue and alkaline fuchsin with the use of recombinant protein turned out to be more efficient (Table 4). In the cases of crystal violet and Coomassie Brilliant blue, there were no significant differences between the native and recombinant laccase. This study confirmed that laccase KbLcc1 from *K. bupleuri* could be a useful biocatalyst for the treatment of dye-containing effluents. Similar results were observed for the laccase from *Phanerochaete flavido-alba*. The recombinant enzyme was characterized by 5× higher specific activity than the native protein. The decolorization reaction of the synthetic dye Remazol brilliant blue R (RBBR) was similar for both variants of laccase. In the case of the dye Acid red 299, the process was much faster for the recombinant enzyme. After 40 min of incubation, over 30% decolorization was observed, with only 13% decolorization for the native laccase. After 100 min, the differences were not so large, approximately 60% and 50% of the dye was removed, respectively, using recombinant and native enzyme [61]. In turn, the native laccase from *P. sanguineus* decolorized the synthetic RBBR dye in 94% within 2 h [62], whereas for the recombinant enzyme, 90% of decolorization was achieved [63].

#### 2.4.2. Biotransformation of Ferulic Acid into Vanillin

In the first stage of the research, the influence of temperature and ferulic acid concentration on the process of obtaining vanillin was determined. Reactions were performed at pH 7.0 at 20 °C and 30 °C, with 1.0 mM and 10.0 mM ferulic acid concentrations. The expected product was found to be present in all tested samples after 5 days. A higher concentration of vanillin was observed when the reaction was carried out at 20 °C. Indeed, it was over 3 × higher than in the reaction at 30 °C (Table 5). The use of more starting material obviously gave rise to more product generation, but the increase is not as great as expected. The yield of the reaction to convert ferulic acid to vanillin is higher when 1 mM substrate is used.

In the next stage of the research, the effect of the redox mediator’s addition was evaluated. The reaction was carried out at 20 °C, using 1 mM ferulic acid as a substrate at pH 7.0 and at pH 3.5, which is optimal for the oxidation of ABTS that acted as a mediator of electron transfer. The concentration of vanillin was determined every 24 h for 5 days. In trials without a mediator, the concentration of vanillin reached the maximum concentration of 1 mg/L. The addition of ABTS and the adjustment of the reaction pH to the mediator resulted in a significant increase in the reaction yield. The highest concentration of vanillin was observed after 96 and 120 h of reaction, at 19.6 and 21.9 mg/L, respectively, with molar yields of 12.85% and 14.39%, respectively (Figure 10 and Appendix A).

Vanillin is the main flavor compound in vanilla extract and probability the most popular universally appreciated global flavor. Concentration of vanillin in vanilla pod reaches 2.5–4.5% of dry mass. Annual global production of vanilla pods, vanilla extract and vanillin is over 18,000,000 kg. Concerning the high production costs of vanilla pods and isolation of vanillin, the industry focuses on the chemical synthesis and biosynthesis of the main flavor ingredient. Chemical synthesis of vanillin has a negative environmental impact due to the use of hazardous chemicals. Biochemical and biotechnological processes are based on the usage of microorganisms and enzymes for transforming a wide range of different substrates into vanillin [22]. The bioconversion of ferulic acid is one of the most investigated ways of obtaining the natural vanillin. Four major pathways of ferulic acid degradation can be distinguished with respect to the initial reaction, namely non-oxidative decarboxylation, side chain reduction, coenzyme-A-independent deacetylation, and coenzyme-A-dependent deacetylation [64]. However, Wellington [65] mentioned research with the use of *T. versicolor* in which ferulic acid was used as a model in a reaction for the oxidative decomposition of lignin by laccase [65]. *P. cinnabarinus* laccase-catalyzed reaction of oxidative decomposition of ferulic acid to vanillin was also described by Falconnier et al. [27]. The transformation process took place during the cultivation of the fungus in a medium containing ferulic acid and resulted with 27.5 molar yield of vanillin bioconversion. More detailed studies, in which the purified enzyme from *P. cinnabarinus* was used for the reaction, did not confirm the previously obtained results, as no formation of vanillin from ferulic acid was observed. The participation of laccase in the described process has not been confirmed, but other enzymes possibly involved in the metabolism of ferulic acid have been identified [27]. In turn, the laccase from *Myceliophthora thermophila* catalyzed the oxidation of ferulic acid in which dye compounds were obtained. Reaction conducted in biphasic hydro-organic system composed of ethyl acetate and sodium phosphate buffer resulted in intermediate stable yellow products which could be applied as food colorants. The authors suggested that they arose as a result of combining the ferulic acid radicals formed after the action of laccase [21]. The results described in this study, observed for the KbLcc1 laccase, are the first to demonstrate the ability of the laccase-like enzyme to produce vanillin from ferulic acid. The processes of such biotransformation described so far concern, inter alia, a coenzyme-independent decarboxylase/oxygenase two-stage process [28]. The first step involved a 2-h decarboxylation reaction of ferulic acid to 4-vinylguaiacol catalyzed by Fdc decarboxylase expressed by recombinant *E. coli*. In the second stage, the recombinant *E. coli* cells expressing oxygenase Cso2 oxidized 4-vinylguaiacol to produce vanillin. The efficiency of the process reached 7.8 g/L of vanillin with 69.3% molar yield [28]. Other methods were described by Perez-Rodriguez with coworkers [26] and Chen with coworkers [25]. In the first mentioned study vanillin production was conducted using bacteria *Amycolatopsis* sp. ATCC 39116. The maximal yield of product was 212.4 mg/L with only 7.37% molar yield after 10 days of fed-batch fermentation in mineral salts medium supplemented with vanillic acid. During the process, other phenolic compounds (vanillic acid, vanillyl alcohol, 4-vinylguaiacol, guaiacol, and protocatechuic acid) were also evaluated [26]. In the second study, vanillin was obtained using bacteria *B. subtilis* which was immobilized onto carbon fiber textiles (CFT) in a bioreactor. During fermentation 1.55 g/L of ferulic acid was used as a sole carbon source for bacterial growth. After 20 h, the vanillin molar yield was 56.59% [25]. Another example is a study of Tang and Hassan [24], who evaluated the production of vanillin by a fungus *A. niger* I–1472. Ferulic acid, a substrate for biotransformation, was initially extracted from pineapple peel and pineapple crown leaves and then converted to 2.5–5.0 mg/L of vanillin with 6.3–8.9% molar yield. It is suggested that ferulic acid–CoA ligase and 4-hydroxycinnamoyl-CoA hydratases/lyase are involved in this process [24].

The new KbLcc1 laccase offers an interesting alternative to obtain vanillin from ferulic acid based on an environmentally friendly bioprocess. The use of a biocatalyst in the production of a value-added product allows for mild reaction conditions and does not generate an excessive amount of waste products. In addition, biotransformation using a single enzyme, *K. bupleuri* G3 IBMP laccase, as opposed to enzyme complexes catalyzing the reactions, are simpler to optimize and more economical. The psychrophilic nature of the enzyme which is efficiently produced at moderate temperatures and does not require heating of bioreactors, also has a significant impact on reducing the costs. It is worth noting that the high thermolability of cold-loving laccase facilitates its quick inactivation after the completed process, thus reducing the risk of the release of the active enzyme into the environment with sewage. Moreover, a single enzyme is easier to immobilize and thus increases the stability of the preparation and enables the biotransformation to be carried out several times using the same portion of the enzyme.

## 3. Materials and Methods

### 3.1. Strains, Plasmids and Media

*K. bupleuri* G3 IBMiP was found to be a new cold-adapted laccase producer [29]. The strain was cultured in the medium containing 10.0 g/L yeast extract, 20.0 g/L peptone, and 20.0 g/L glucose for 3 days at 20 °C. The *E. coli* strain Top10 F’ (ThermoFisher Scientific, Waltham, MA, USA) were grown in a Luria-Bertani medium (LB) at 37 °C. The *E. coli* transformants were grown in the LB medium with 100 µg/mL of ampicillin at 37 °C. pJET1.2/blunt Cloning Vector (ThermoFisher Scientific, Waltham, MA, USA) was used for splicing in vitro and subcloning of DNA fragments for sequencing. The vector pPINKα-HF (Invitrogen, Carlsbad, CA, USA) was used for the expression in *P. pastoris* (PichiaPink^TM^ strain 1). Selective medium (SM, in g/L: glucose, 20.0; yeast nitrogen base without amino acids, 6.7; yeast synthetic drop-out medium supplements without histidine, leucine, tryptophan and adenine, 1.39; bacteriological agar, 20.0; histidine, 0.076; leucine, 0.38; tryptophan, 0.076), buffered complex glycerol and methanol media (BMGY and BMMY, in g/L: yeast extract, 10.0; peptone, 20.0; potassium phosphate, 100 mM pH 6.0; yeast nitrogen base, 13.4; biotin, 0.004; glycerol, 10.0, or methanol, 5.0) and other media and protocols for *P. pastoris* were as described in the PichiaPink^TM^ Expression Manual (Invitrogen, Carlsbad, CA, USA) unless otherwise stated.

### 3.2. Cloning of Laccase Gene from K. bupleuri G3 IBMiP

#### 3.2.1. Genome Sequencing

The laccase-encoding gene was identified in the *K. bupleuri* G3 IBMiP genome sequence. Sequencing was performed in the Bioinformatics Laboratory at Bionanopark in Łódź, Poland. Libraries were prepared according to the NExter DNA protocol (Ilumina, San Diego, CA, USA) and pair-end sequenced on a MiSeq device (Ilumina, San Diego, CA, USA). The SPAdes programme (version 3.8.1) was used to assemble the obtained data into a genome sequence (http://bioinf.spbau.ru/spades, accessed on 14 July 2016) [66].

#### 3.2.2. Bioinformatics Analysis of Laccase Gene

Genes encoding for laccases were identified on the basis of annotations carried out using the programmes: Augustus (http://augustus.gobics.de/, accessed on 1 August 2016) and GeneMark (http://exon.gatech.edu/GeneMark/, accessed on 15 July 2016). The search of the obtained sequences for domains characteristic of multicopper oxidases was performed using Pfam 31.0 database (http://pfam.xfam.org/, accessed on 15 July 2016) and in the Blast Standalone programme against the CDD database (https://blast.ncbi.nlm.nih.gov/Blast.cgi, accessed on 18 July 2016). Selected sequences encoding laccase-like proteins were analyzed for the presence of introns and the characteristics of the encoded protein (molecular weight, signal peptide, pI, etc.) using the tools available on the Expasy server (https://www.expasy.org/, accessed on 18 August 2016).

#### 3.2.3. KbLcc1 Gene Amplification and Cloning

The laccase gene KbLcc1 with the intron sequence was amplified using primers L1 LF/L1 LR (Table 6) with genomic DNA as the template and using a combined 2-step and 3-step touch-down PCR protocol (Table 7). The amplified product was purified using GeneMatrix Basic DNA Purification Kit (EURx, Gdansk, Poland) and cloned to pJET1.2/blunt Cloning Vector using CloneJET PCR Cloning Kit (ThermoFisher Scientific, Waltham, MA, USA).

#### 3.2.4. Splicing In Vitro of KbLcc1 Gene

Coding sequence of laccase KbLcc1 was obtained using the OE-PCR method according to the protocol described by Heckman and Pease [30] and presented at Figure 2. First PCR reactions were performed using primers L1 LF/IntR to amplify exon 1 (product AB) and IntF/L1 LR to amplify exon 2 (product CD), with the conditions presented in Table 7 and the plasmid pJET1.2 with cloned KbLcc1 gene sequence used as a template. Products AB and CD were extracted from the agarose gel using agarose-out protocol from the GeneMatrix Basic DNA Purification Kit (EURx, Gdansk, Poland) and then used as the templates for PCR with primers L1 SF/L1 SR (Table 6) to obtain product AD (coding sequence of KbLcc1 laccase). Reaction conditions of PCR steps included the initial denaturation at 98 °C for 30 s; 25 cycles at 98 °C for 10 s denaturing, annealing at 64.5 °C for 30 s, and extension at 72 °C for 60 s; a final extension of 72 °C for 10 min followed by maintenance at 4 °C. Gel-purified product AD was cloned to pJET1.2/blunt Cloning Vector using CloneJET PCR Cloning Kit (ThermoFisher Scientific, Waltham, MA, USA) and the obtained vector was transformed into *E. coli* Top10 F’ and plated on LB medium with ampicillin (100 µg/mL).

#### 3.2.5. Cloning of KbLcc1 Coding Sequence to Expression Vector

Plasmid pJET1.2 with KbLcc1 coding sequence was isolated using GeneMatrix Basic DNA Purification kit (EurX, Gdansk, Poland) and used as a template in PCR reaction to add StuI and FseI restriction sites at the beginning and the end of coding KbLcc1 sequence using primers L1 StuIF/L1 FseIR. Reaction conditions of PCR steps included the initial denaturation at 98 °C for 30 s; 25 cycles at 98 °C for 10 s denaturing, annealing at 65.9 °C for 30 s, and extension at 72 °C for 60 s. A final extension of 72 °C for 10 min was followed by maintenance at 4 °C. Purified product was cloned to pJET1.2/blunt Cloning Vector using CloneJET PCR Cloning Kit (ThermoFisher Scientific, Waltham, MA, USA) and the obtained vector was transformed into *E. coli* Top10 F’ and plated on LB medium with ampicillin (100 µg/mL). Isolated and purified plasmid pJET1.2 with KbLcc1 coding sequence and plasmid pPINKα-HF were digested by FastDigest StuI (NEB, Ipswich, MA, USA) and FastDigest FseI (NEB, Ipswich, MA, USA) for 30 min at 37 °C. Gel-purified pPINKα-HF vector and KbLcc1 gene were ligated by Anza T4 DNA Ligase (Promega, Madison, WI, USA) at 22 °C for 1 h, and the product was used for electroporation of *E. coli* Top10 F’ at 1.8 kV with a 0.1-cm cuvette according to the protocol attached to MicroPulser (Bio-Rad, Hercules, CA, USA). The obtained construct was subjected to multi-enzyme digestion and linearization to confirm the correct length.

#### 3.2.6. Expression of the KbLcc1 Laccase in *P. pastoris*

The recombinant plasmid pPINKα-HF containing laccase KbLcc1 sequence was linearized with FastDigest SpeI according to the manufacturer’s instruction (NEB, Ipswich, MA, USA) and transformed into *P. pastoris* (PichiaPink^TM^ strain 1) competent cells by electroporation (MicroPulser, Bio-Rad, Hercules, CA, USA) by 1 pulse at 2.0 kV for 5 msec with a 0.2-cm cuvette. The transformants were screened on SM plates. Positive clones were identified by colony PCR with the primers 5′AOX1/3′CYC1 (Table 6).

Twenty-three positive clones of *P. pastoris* bearing the KbLcc1 gene, and one “empty” clone without gene, were incubated in 60 mL of BGMY for 48 h (24 °C, 225 rpm). Cells were collected by centrifugation and resuspended in 6 mL of BMMY containing 4.0% (*v*/*v*, final concentration) of methanol, which was added every 24 h to induce the expression of the target protein. Cultivation was conducted at 20 °C for 5 days with shaking (225 rpm). The supernatant was collected by centrifugation at 5000× *g* for 5 min at 4 °C every 24 h during incubation time, and laccase activity and protein concentration were determined. SDS-PAGE was performed in a 12% running gel, and proteins were stained with Coomassie Brilliant Blue R-250. KbLcc1 laccase production for characterization of the recombinant protein was conducted with clone CI of *P. pastoris* at the same conditions, but with 2 mM CuSO_4_ and 0.5% methanol concentration in BMMY medium and in larger volume scale. Copper ions were added to sterile BMMY medium.

#### 3.2.7. Purification of Recombinant Laccase KbLcc1

The supernatant from 1 L culture of the recombinant *P. pastoris* clone CI was harvested by centrifugation at 10,000 rpm for 30 min at 4 °C. The supernatant was concentrated by ultrafiltration using Sartorius Stedim system and the membrane with cut-off of 10 kDA (Hydrosart, Sartorius, Göttingen, Germany). The recombinant enzyme was precipitated by ammonium sulfate fractionation. The precipitate was dissolved in 10 mM sodium acetate buffer (pH 5.5) and then dialyzed for 12 h against the same buffer. Next, concentrated and pre-purified protein solution was loaded onto MonoS cation exchange column (GE Healthcare, Chicago, IL, USA) connected to ÄKTA Purifier system. The bound protein was eluted by NaCl gradient in 10 mM sodium acetate buffer (pH 5.5). The flow rate was set to 1 mL/min.

### 3.3. Characterization of Purified Recombinant Laccase KbLcc1

#### 3.3.1. Effect of Temperature and pH on Laccase Activity and Stability

The optimum temperature for laccase activity was determined by incubating the reaction mixture over a temperature range of 0–80 °C at a pH optimal for substrates (1 mM, 30 µM, 2 mM and 250 µM concentration of ABTS, syringaldazine, guaiacol and sinapic was used). To investigate the thermostability of the enzyme, it was incubated at 5–80 °C for 0–120 min within a 30 min interval, followed by measuring the enzyme activity at pH 3.5 with 1 mM ABTS as a substrate. The effect of pH on laccase activity for different substrates was determined in 10 mM Britton-Robinson buffer at pH in the range of 2.0–13.5 used in the reaction mixture. To investigate the pH stability, the enzyme (1.0 U) was incubated for 24 h at 4 °C in 10 mM Britton-Robinson buffer at pH in the range of 2.0–12.0 and then 0.1 U of the enzyme after incubation was used to determine the relative activity of laccase. The activity was determined with ABTS as a substrate at the optimum temperature (30 °C) and optimal pH from ABTS (3.5). All the reactions were performed in triplicate. All experiments were conducted with 0.1 U of the purified recombinant laccase KbLcc1.

#### 3.3.2. Substrate Specificity of Recombinant Laccase

Substrate specificity of laccase KbLcc1 from *K. bupleuri* was estimated using 4 substrates: ABTS (ε_420_ = 3.6 × 10^4^ M^−1^ cm^−1^; conc. 0.2–4.0 mM), syringaldazine (ε_526_ = 6.5 × 10^4^ M^−1^ cm^−1^; conc. 5.5–40.0 µM), guaiacol (ε_526_ = 4.8 × 10^4^ M^−1^ cm^−1^; conc. 2.5–20.0 mM) and sinapic acid (ε_312_ = 1.76 × 10^4^ M^−1^ cm^−1^; conc. 3.0–150 µM). Firstly, the optimum pH for each substrate was established in 10 mM Britton-Robinson buffer pH 2.0–10.0 and with 1 mM, 30 µM, 2 mM, and 250 µM concentration of ABTS, syringaldazine, guaiacol and sinapic acid used as a substrate, respectively. Activity determination was carried out at 30 °C. Kinetic parameters, namely K_M_, V_max_, and k_cat_, were determined using the Lineweaver-Burk plots.

#### 3.3.3. The Effects of Metal Cations and Chemicals on the Activity and Stability of the Laccase

The laccase activity was determined in the presence of various metal ions (Mg^2+^, Ca^2+^, Ni^2+^, Co^2+^, Zn^2+^, Cu^2+^, Mn^2+^, Fe^2+^, Fe^3+^). The metal ions were added to the reaction mixture at 1 mM, 2 mM, 5 mM, or 10 mM concentration. Laccase activity was assayed with ABTS as a substrate under standard conditions described below. The stability of laccase in the presence of the same metal ions was studied by preincubating the buffered enzyme for 24 h and 10 days at 4 °C followed by enzyme activity assay. The metal ions were added to the enzyme in acetate buffer (100 mM, pH 3.5) to a final concentration of 1.0, 2.0, and 5.0 mM. After incubation, samples were diluted to minimize the concentration of metal ions in the reaction mixture. A control sample preincubated without the metal ions was also performed. All assays were carried out in triplicate with 0.1 U of the purified recombinant laccase KbLcc1.

The laccase activity was determined in the presence of various organic solvents (acetone, ethyl acetate, methanol, ethanol, isopropanol, hexane). The organic solvents were added to the reaction mixture at 1%, 10%, 20%, and 40% (*v*/*v*) concentrations. Laccase activity was assayed with ABTS as a substrate under standard conditions described below. The stability of laccase in the presence of the same organic solvents were studied by preincubating the enzyme at 4 °C for 24 h followed by enzyme activity assay. The organic solvents were added to the enzyme in acetate buffer (100 mM, pH 3.5) to a final concentration of 1%, 10%, 20%, and 40% (*v*/*v*). After incubation, samples were diluted with buffer to minimize the concentration of organic solvents in the reaction mixture. A control sample preincubated without organic solvent was also performed. All assays were carried out in triplicate with 0.1 U of the purified recombinant laccase KbLcc1.

### 3.4. Application of Laccase KbLcc1

#### 3.4.1. Decolorization of Synthetic Dyes

Decolorization of four synthetic dyes: methylene blue, alkaline fuchsin, Coomassie Brilliant Blue, and crystal violet, was performed with the purified KbLcc1 laccase according to our previously described method [29]. The reaction mixture contained acetate buffer (100 mM, pH 3.5), synthetic dye (250 mg/L), purified laccase KbLcc1 (0.1 U), and 1.0 mM ABTS as a redox mediator. The mixture was incubated at 30 °C for 60 min. The decolorization of dye, expressed as dye decolorization percent (%), was calculated by means of Formula (1):decolorization (%) = [(A_i_ − A_f_)/A_i_] × 100,(1)
where A_i_ is the initial concentration of the dye and A_f_ the final concentration of the dye. All reactions were performed in triplicates.

#### 3.4.2. Biotransformation of Ferulic Acid into Vanillin

The recombinant KbLcc1 enzyme was used in a biotransformation of ferulic acid into vanillin. The reactions were carried out under various conditions. The reaction mixture contained phosphate buffer (100 mM, pH 7.0) or acetate buffer (100 mM, pH 3.5), ferulic acid as a substrate (in the final concentrations of 1 mM or 10 mM), 0.1 U of the purified recombinant laccase KbLcc1, and no or 1 mM of ABTS added as redox mediator. The mixture was incubated at 20 °C or 30 °C for 5 days. The reaction products were extracted with methylene chloride (1:1, *v*/*v*) and organic phase was analyzed by gas chromatography coupled to mass spectrometry (GC-MS). Molar yield was calculated as the ratio between the produced vanillin (mM) and the initial ferulic acid (mM).

### 3.5. Analytical Methods

Laccase activity was determined by analysis of the oxidation of ABTS [67]. The nonphenolic dye ABTS is oxidized by laccase to the more stable and preferred state of the cation radical. The concentration of the cation radical responsible for the intense blue-green color can be correlated to the enzyme activity and is measured at 420 nm [68] at 30 °C. The assay mixture contained 1.0 mM ABTS, 100 mM acetate buffer (pH 3.5), and a suitable amount of enzyme. Oxidation of ABTS was monitored by determining the increase in A_420_ (ε_420_, 3.6 × 10^4^ M^−1^ cm^−1^) for 10 min. One unit (U) was defined as the amount of the laccase that oxidized 1 μmol of ABTS substrate per min.

Protein concentration was determined according to Bradford [69] using BSA as a standard.

Determination of vanillin concentration was performed using the GC-MS spectrometer Pegasus 4 D (LECO, St. Joseph, MI, USA). The chromatography column used was Stabilwax-DA (Restek, Bellefonte, PA, USA) (length of the column 30 m, inner diameter 0.25 mm, thickness of the film of the stationary phase 0.25 µm). The temperature of the injector was equal to the temperature of the transfer lane (250 °C). The temperature program of the column oven was set from 50 °C (isotherm 1 min) to 240 °C (isotherm 15 min) with a temperature rise of 4 °C/min. Helium gas was used as a carrier gas with the constant flow of 1 mL/min. The mass spectrometer was set to Electron Impact mode, with the temperature of the ion source equal to 200 °C, and the energy of ionization 70 eV.

### 3.6. Statistical Analysis

Calculations were performed using Microsoft Excel version 2007 (Microsoft Corporation, Redmond, WA, USA). Experimental values were reported as the means ± s.e. Statistical significance of the data was assessed by analysis of variance (*p* < 0.05) using the MiniTab19 software (MiniTab Lt, Coventry, UK, Software; www.minitab.com/en-us/products/minitab/, accessed on 10 June 2021). For significant differences, the Tukey method of multiple comparisons was performed. Graphs were plotted using SigmaPlot version 11 (Chicago, IL, USA) and Graph Pad Prism5 (version 5.00, San Diego, CA, USA).

## 4. Conclusions

The conducted research reported the identification, splicing in vitro, and heterologous expression of the new laccase KbLcc1 gene from *K. bupleuri* G3 IBMiP in *P. pastoris*. KbLcc1 laccase possesses interesting enzymatic properties (low optimal temperature, broad pH range, and organic solvent tolerance) which predetermine its application in organic synthesis reactions. Moreover, increased stability in the presence of Ca^2+^ and Mg^2+^ ions could be an advantage in processes taking several days and in the long-term storage of the enzyme. The recombinant laccase was proven to be more efficient in the decolorization of the synthetic dyes than its native counterpart, reaching almost 50% yields (for crystal violet). It is worth emphasizing that, so far, no application of laccases in the biotransformation of vanillin has been described in the literature. For the first time, this research presented the results of the bioconversion of ferulic acid to vanillin by the recombinant KbLcc1 laccase. This process requires further studies, especially in terms of the reaction mechanism. It is necessary to conduct more detailed experiments aimed at tracing the intermediate and side products formed in the reaction.

## Figures and Tables

**Figure 1 ijms-22-09593-f001:**
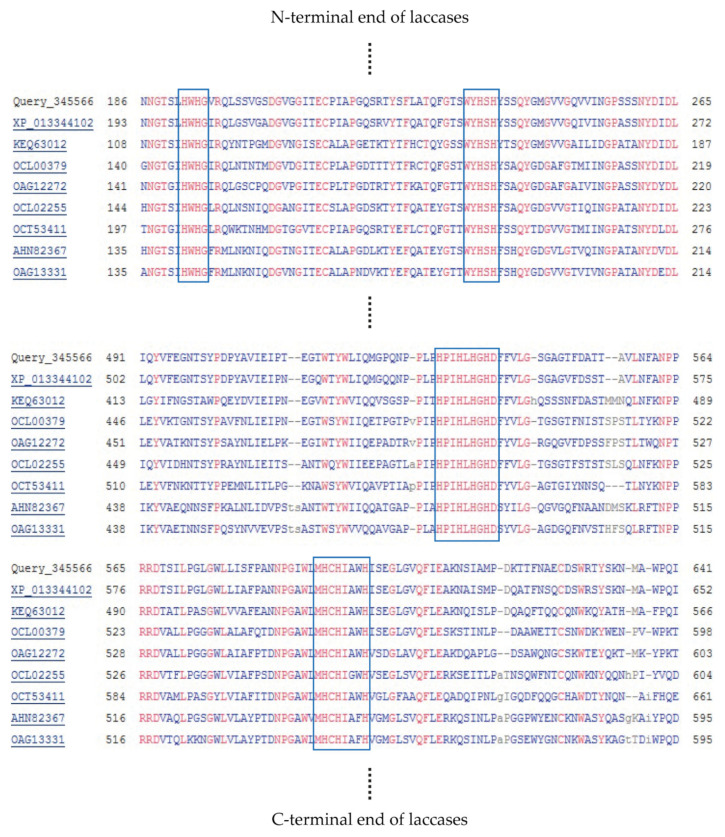
Multiple sequence alingment of laccase KbLcc1 and other laccase sequences from *Ascomycetes* using Constraint-based Multiple Alignment Tool (COBALT, https://www.ncbi.nlm.nih.gov/tools/cobalt/cobalt.cgi, accessed on 15 July 2021); strong conservative copper-binding amino acids are indicated by blue frames.

**Figure 2 ijms-22-09593-f002:**
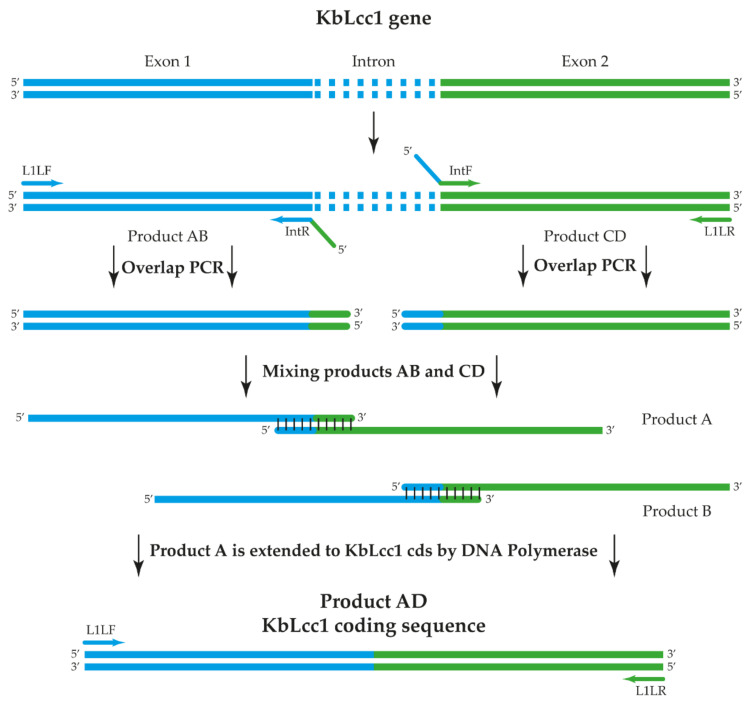
Scheme of Overall-Extension PCR reaction used to obtain of KbLcc1 laccase coding sequence.

**Figure 3 ijms-22-09593-f003:**
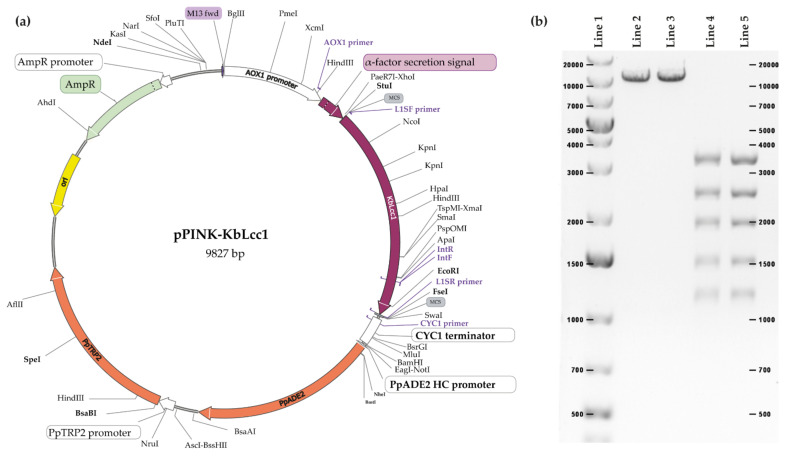
Construct of pPinkα-HF vector with the insert of laccase KbLcc1 coding sequence (**a**) and its restriction analysis (**b**); DNA fragments length marker (line 1), product after SpeI enzyme treatment (lines 2 and 3), products after multi-enzymes digestion with SpeI, AscI, NdeI and EcoRI (lines 4 and 5).

**Figure 4 ijms-22-09593-f004:**
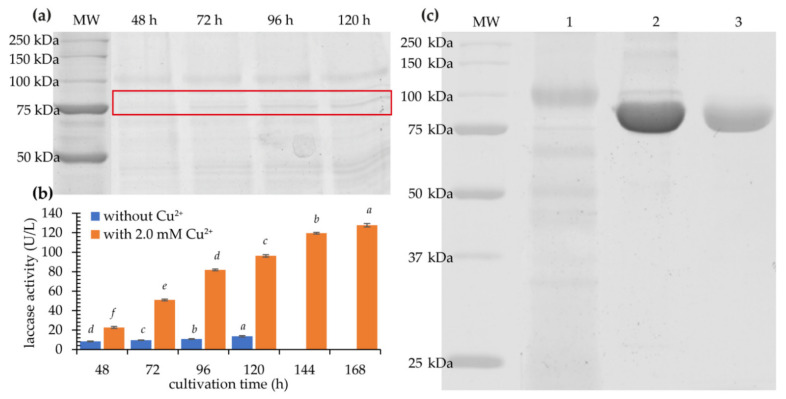
SDS-PAGE of the recombinant protein; (**a**) laccase KbLcc1 production after 48, 72, 96 and 120 h cultivation of *P. pastoris* in expression medium BMMY (without Cu^2+^); KbLcc1 laccase bands were indicated by the red frame; (**b**) increase of laccase KbLcc1 activity during cultivation time in medium without and with copper supplementation (2.0 mM Cu^2+^); the data presented as the means ± SD from at least three independent experiments; the means that are marked with different letters (*a*–*d* for data series without Cu^2+^ and *a*–*f* for data series with 2.0 mM Cu^2+^) are significantly different (*p* < 0.05); (**c**) purification of KbLcc1, 1: ultrafiltrated crude post-culture supernatant after 168 h of expression with 2.0 mM Cu^2+^; 2: fraction after salting-out using ammonium sulfate; 3: purified KbLcc1; MW: molecular weight marker.

**Figure 5 ijms-22-09593-f005:**
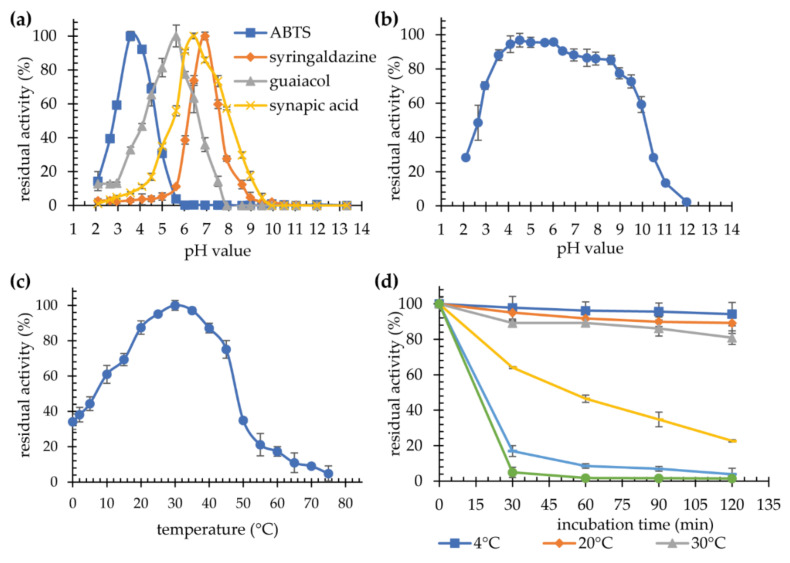
Characterization of purified recombinant laccase KbLcc1; (**a**) optimal pH in reaction with different substrates; (**b**) pH-stability after 24-h incubation at 4 °C (with ABTS used as a substrate); (**c**) optimal temperature in reaction with ABTS as a substrate; (**d**) thermostability at 4 °C, 20 °C, 30 °C, 40 °C, 45 °C and 50 °C; pH-stability and thermostability obtained with ABTS as a reaction substrate.

**Figure 6 ijms-22-09593-f006:**
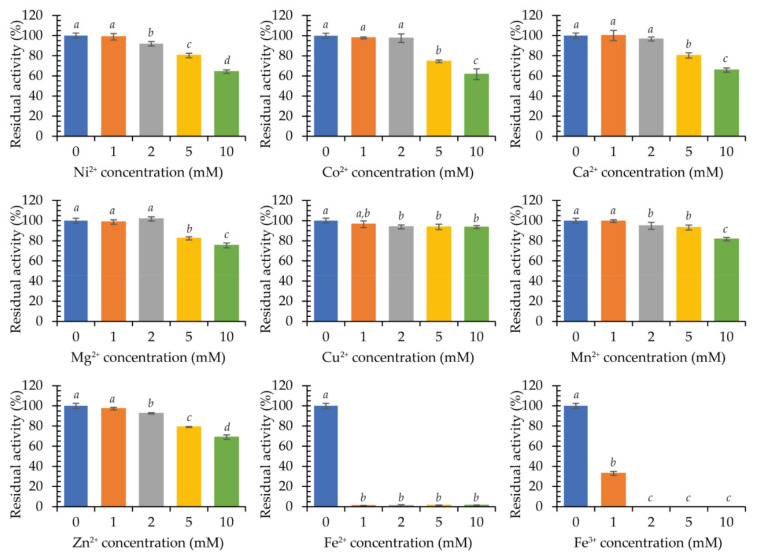
Activity of purified recombinant laccase KbLcc1 in the presence of various metal ions at 1 mM, 2 mM, 5 mM and 10 mM concentration in the reaction mixture; the data presented as the means ± SD from at least three independent experiments; the means that are marked with different letters (*a*–*d*) are significantly different (*p* < 0.05).

**Figure 7 ijms-22-09593-f007:**
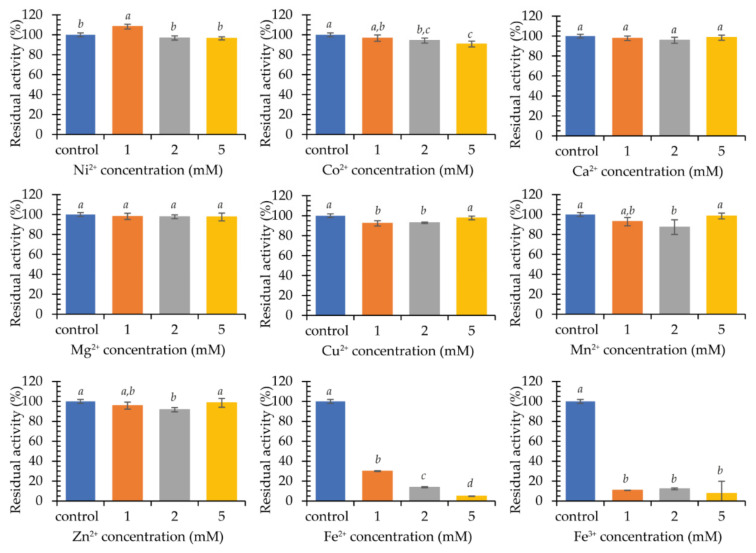
Stability of purified recombinant laccase KbLcc1 activity after 24 h of pre-incubation in the presence of various metal ions; the data presented as the means ± SD from at least three independent experiments; the means that are marked with different letters (*a*–*d*) are significantly different (*p* < 0.05).

**Figure 8 ijms-22-09593-f008:**
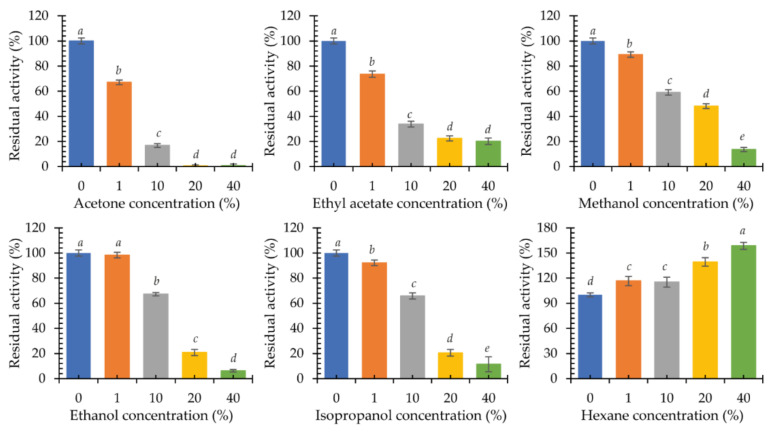
Activity of purified recombinant laccase KbLcc1 in the presence of organic solvents at 0%, 1%, 10%, 20%, 40% (*v*/*v*) concentration; the data presented as the means ± SD from at least three independent experiments; the means that are marked with different letters (*a*–*e*) are significantly different (*p* < 0.05).

**Figure 9 ijms-22-09593-f009:**
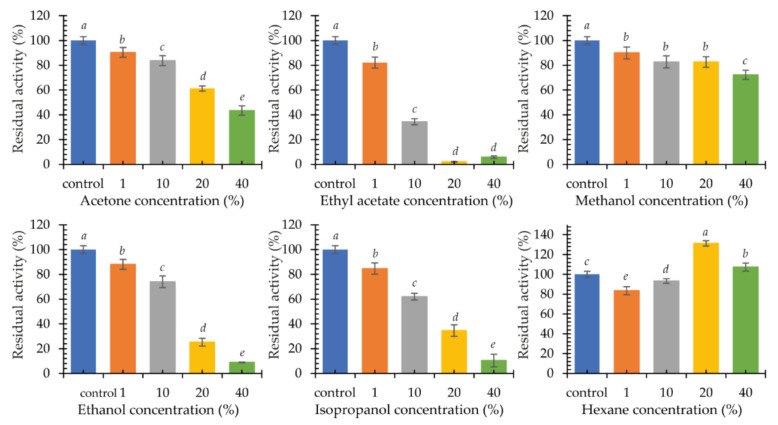
Stability of purified recombinant laccase KbLcc1 activity after 24-h pre-incubation with organic solvents at 1%, 10%, 20% and 40% (*v*/*v*) concentration; the data presented as the means ± SD from at least three independent experiments; the means that are marked with different letters (*a*–*e*) are significantly different (*p* < 0.05).

**Figure 10 ijms-22-09593-f010:**
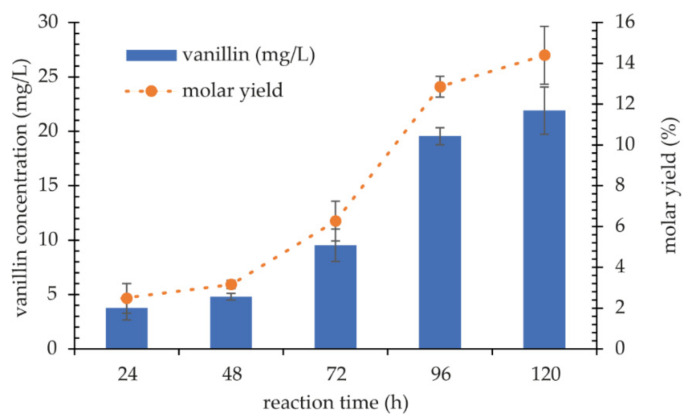
Transformation of the ferulic acid into vanillin by KbLcc1 laccase (0.1 U) with 1 mM ABTS as a redox mediator at 20 °C and pH 3.5 (optimal for ABTS oxidation).

**Table 1 ijms-22-09593-t001:** Accession numbers of sequences of laccases produced by *Ascomycetes* used for multiple sequence alignment.

Organism	GenBank Accession Number
*Kabatiella bupleuri* G3 IBMiP	MZ292708
*Aureobasidium subglaciale* EXF–2481	XP_013344102.1
*Aureobasidium melanogenum* CBS 110374	KEQ63012.1
*Cenococcum geophilum* 1.58	OCL00379.1
*Paraphaeosphaeria sporulosa*	OAG12255.1
*Glonium stellatum*	OCL02255.1
*Cladophialophora carrionii*	OCT53411.1
*Shiraia* sp. SUPER–H168	AHN82367.1
*Alternaria alternata*	OAG13331.1

**Table 2 ijms-22-09593-t002:** Summary of the purification procedure for laccase KbLcc1.

Purification Step	Total Activity (U)	Total Protein (mg)	Specific Activity (U/mg)	Yield (%)	Purification (fold)
Crude extract	125.15	298.90	0.419	100%	1.00
Ultrafiltration	105.26	139.15	0.756	84.11%	1.81
Ammonium sulfate precipitation	65.96	54.24	1.216	52.71%	2.90
Mono S columnion exchange chromatography	36.78	1.795	20.49	29.39%	48.94

**Table 3 ijms-22-09593-t003:** Kinetic parameters for purified recombinant laccase KbLcc1.

Substrate	T_opt_ (°C)	pH_opt_	K_M_ (mM)	V_max_ (U mg^−1^)	k_cat_ * (s^−1^)	k_cat_/K_M_ (mM^−1^ s^−1^)
ABTS	30	3.5	0.580	34.18	39.64	68.35
Syringaldazine	30	7.0	0.021	33.03	38.31	1808.82
Guaiacol	30	5.5	13.28	4.70	5.45	0.41
Sinapic acid	30	6.5	0.015	1.39	1.61	109.26

* The molecular weight of KbLcc1 was 69.6 kDa, which was used to calculate the k_cat_ (s^−1^) values. All of the values were calculated by the linear regression (correlation coefficient ≥ 0.99) of double reciprocal plots, f(1/[S]) = 1/[v], from every set of triplicate measurement (SD < 5%) (Appendix A, Appendix A).

**Table 4 ijms-22-09593-t004:** Decolorization of the synthetic dyes using native laccase G3 [29] and purified recombinant laccase KbLcc1 from *Kabatiella bupleuri* G3 IBMiP expressed in *Pichia pastoris*; reaction mixture contained 0.1 U of laccase, 250 mg/L of synthetic dye and 1 mM of ABTS as a redox mediator; the data presented as the means ± SD from at least three independent experiments; the means that are marked with different letters (*a*–*b*) are significantly different (*p* < 0.05).

Dye	Group	Chemical Structure	% Decolorization	
			Native	Recombinant
Methylene blue	Heterocyclic/thiazine dye	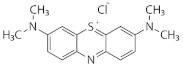	18.2 ± 2.5% *^b^*	23.8 ± 1.2% *^a^*
Alkaline fuchsin	Triphenylmethane dye and aniline dye	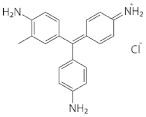	31.7 ± 3.3% *^b^*	36.6 ± 1.8% *^a^*
Crystal violet	Triphenylmethane dye	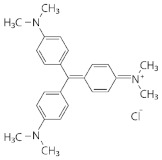	40.4 ± 7.0% *^a^*	48.1 ± 4.3% *^a^*
Coomassie Brilliant Blue R-250	Triphenylmethane dye	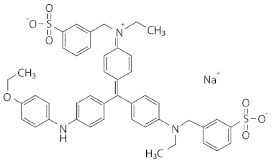	19.8 ± 5.8% *^a^*	24.4 ± 3.9% *^a^*

**Table 5 ijms-22-09593-t005:** Vanillin concentration and molar yield of transformation ferulic acid by purified KbLcc1 laccase (0.1 U) without the redox mediator.

Ferulic Acid (mM)	20 °C		30 °C	
	Vanillin (mg/L)	Molar Yield (%)	Vanillin (mg/L)	Molar Yield (%)
1.0	0.136	0.089	0.082	0.054
10.0	0.291	0.019	0.119	0.008

**Table 6 ijms-22-09593-t006:** Primers used in PCR reactions.

Name of Primer	Sequence (5′–3′)
L1 LF	*ATGCATCTACAGAAGCTCAGTGGCCTCT*
L1 LR	*CTAAAGTCCACTGTCGATTTGAGGCCAAGC*
IntR	*AAATCGTGACCGTGCAAATGGATGGGATGA*
IntF	*CATTTGCACGGTCACGATTTCTTCGTCCTC*
L1 SF	*ATGCATCTACAGAAGCTCAGTG*
L1 SR	*CTAAAGTCCACTGTCGATTTGAG*
L1 StuIF	*AGGCCTATGCATCTACAGAAGCTC*
L1 FseIR	*GGCCGGCCTAAAGTCCACT*
5′AOX1	*GACTGGTTCCAATTGACAAGC*
3′CYC1	*GCGTGAATGTAAGCGTGAC*

**Table 7 ijms-22-09593-t007:** Combined 2-step and 3-step touch-down PCR protocol.

Cycle Step	Temperature (°C)	Time	Cycles
Initial denaturation	98	30 s	
Denaturation	98	10 s	5
Annealing/Extension	72	60 s
Extension	72	5 min	
Denaturation	98	10 s	20
Annealing	72 (−1/cycle)	30 s
Extention	72	60 s
Final extension	72	10 min	
Cooling	4	hold

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
