# Peer review of "Novel Cold-Adapted Recombinant Laccase KbLcc1 from Kabatiella bupleuri G3 IBMiP as a Green Catalyst in Biotransformation"

_ijms, 2021, doi:10.3390/ijms22179593_

Round 1

Reviewer 1 Report

The article by Wisniewska et al entitled “Novel cold-adapted recombinant laccase KbLcc1 from Kabatiella bupleuri G3 IBMiP as a green catalyst in biotransformation” shows the cloning, expression, purification and functional characterization of an enzyme of potential industrial interest. The experiments are well performed and the sequence is neat and the results are very interesting, especially those related to vanillin synthesis.

However, the presentation of the results throughout the text is chaotic and the level of English is low, including basic grammatical errors. Therefore, revision by an English grammar editor is necessary. In addition, in my opinion, figures are missing, with some data appearing in the text that seem incomplete and that with a figure showing the experiment would undoubtedly be clearer. The figure captions are undoubtedly incomplete and the explanation of the results is scarce and sometimes unconnected between figures, which make it difficult to follow the thread of the article. It gives the sensation that it is not written for the general public and only for a more specialized public in the matter.

Finally, I find that although the results shown in the article are of considerable interest, the discussion of them is limited to comparing them with existing data without making any effort to highlight the improvements that the enzyme shown here has with respect to similar ones already published. This is especially noticeable in the last part of the discussion.

Major points

  • English grammar revision needed.

  • In no case is it indicated whether the assays are performed with purified protein or with fractions of the supernatant from the p. pastoris expression. Thus, for example, in lines 143-153 the expression and activity are mentioned without detailing whether the experiments have been performed with purified protein.

  • Line 206. Maximum activity was recorded at 30ºC for all substrates. Where is this basic information? Please add a Figure showing substrates activities at different temperatures. Specify for all the experiments, substrate concentration and reaction times, including this information both in text and figure captions.

  • Both in the text and in the Figures there is a lack of information regarding the concentrations of substrates used or the reaction times for example. In the specific case of the effect of copper ions on the enzyme activity, the results, at least what is apparently deduced from the text, seem to be different when talking about the overexpression in pastoris and the characterization of the purified protein. In this particular case, a more detailed explanation of the experiments (were they performed with purified protein or with extracts of the supernatant?) and the addition of new Figures that help to follow the results would help in their understanding.

  • Undoubtedly the most interesting result, from my point of view, is the one related to the ability of the enzyme to catalyze the synthesis of vanillin from ferulic acid. However, the experiments performed are scarce and should undoubtedly be reinforced with more experimental observations. The identification of the optimum temperature for this reaction, as well as its optimum pH (not showing only two pH points, but as shown in Figure 5a), optimum substrate concentration (kinetic characterization) and optimum ABTS concentration, are certainly necessary results to meet the quality standards of the journal.

Minor points

  • Lines 34 and 37. Change “process” to “processes”

  • Homogenize microorganisms nomenclature. I understand that for the most know species such as coli or P. pastoris, the whole name is not necessary, but for the rest it would be great to show the whole name. As an example, in line 70, all but one (A. niger) are named with the whole scientific name. Please check this throughout the text.

  • Line 86. Is K. bupleuri a yeast? Or is a yeast-like fungus?

  • Line 93. Make sentence shorter. I would propose the following. “contained a domain characteristic for laccases. Only three of them…”

  • Line 93. Which domain? Please specify.

  • Line 93. Change the hyphen by “contained”

  • Line 94. How was kbLcc1 selected out of the three final candidates?

  • Lines 98-101. Not clear at all. How is it possible to get such a different results only by using two different databases? Was the sequence coverage for the resulting sequences identities different? crassa resulting protein sequence is not referred again throughout the text, why then you mentioned it here? Not even discussed.

  • Figure 1 legend. Add the programme name used to perform sequence alignment.

  • Lines 144-146. Time units should be expressed homogeneously. 5 days is equivalent to 120 hours.

  • Lines 144 and 153. Is the activity at 5 and 7 days exactly the same (13.6 U/L)?

  • Lines 142-153. A Figure showing protein activity until day 7 with and without Cu2+ should be shown.

  • What does “2.5 mM turned out to be too high” mean? Protein stops working?

  • Lines 142-153. Data regarding Cu2+ effects don't match those shown in Figures 6 and 7. A Figure showing results from lines 142-153 and a clearer explanation about the experimental conditions and results should be added.

  • Line 157. Change CuSO4 to CuSO4

  • Substitute “approx.” throughout the text.

  • Line 169. “7-days”. Remove hyphen and express in days or hours based on what was mentioned above.

  • Line 175. lccA to LccA.

  • Lines 181. Figure 4 to Figure 4b

  • Table 2. Any idea where is the protein lost during MonoS purification? Not properly eluted? Unfolded during purification protocol? Not attached to the column?

  • Line 183. Run a gel after endoglycosidase treatment to confirm the glycosilated nature of your protein.

  • Figure 4. A gel including a 7 days expression point should be added.

  • Figure 4 Legend. Remove “MW: molecular marker” and move “MW: molecular weight marker” at the end of the caption.

  • Line 202. From here on, do I have to imagine that the protein characterization has been performed with purified protein? If so, please specify at the beginning of the corresponding sections.

  • Lines 208-209. Regarding enzyme stability at a wide range of pHs for 24h (Fig 5b), which substrate was used to measure enzyme activity? Only ABTS? One pH point for each substrate? Please specify both in the text and in the Figure legend.

  • Lines 224-225. Change Table3 to Table 3.

  • Line 225. Change “Consistent with the result for the native…” to “ Consistent with that reported for the native…”

  • Figure 5c. Which substrate was used? Concentration and reaction time? Add information to both text and Figure legend.

  • Figure 5c. Is thermal stability identical for all the substrates? Please show that information.

  • Line 256. Change 521mM to 0.521 mM.

  • Lines 275-285. Difficult to follow. If I understood correctly, Figure 6 represents ion effect on enzyme activity in a “real time” experiment, while Figures 7 and S1 represent cation effect after 24h or 10 days incubation, respectively. Is that right? If so, please specify that Figure 6 experiments were conducted without pre-incubation of the protein with the cations but they were present in the reaction medium.

  • Lines 308-309. How long the enzyme produced by S. commune retained activity in the presence of Cu2+?

  • What does a, b, c, and d mean? Add this information to Figure Legends.

  • Lines 465-487. Improve Discussion. Not clear the final message.

  • Line 501. Once protein purified, which protein concentration you reached? Is the purified protein stored under -80ºC, 4ºC, RT….?

  • Lines 620-622. Add a Figure showing the kinetic characterization of the enzyme.

Reviewer 2 Report

This is a detailed and well-written paper describing the heterologous expression of a laccase (KbLcc1) that the same group identified previously in the yeast Kabatiella bupleuri.  Many laccases are known, which limits the novelty, but the authors describe the application of the KbLcc1 to the production of vanillin from ferulic acid, which is very interesting.  The manuscript could be improved with some changes:

  1. Line 93.  This is confusing.  How did the authors identify the laccase they eventually expressed?  There were three that had the necessary domains and signal sequences.
  2. Figure 2 is made difficult to interpret with the large text claiming 'no Cu2+ bonding sites identified' and a legend stating that 'strong conservative copper-binding amino acids are indicated by blue frames'.  I suggest removing the text on the figure.
  3. Line 173 describes the laccase as psychrophilic, meaning that it prefers lower temperatures.  However, there is no evidence for this; the enzyme's temperature optimum is 30 oC, which is mesophilic.
  4. Figure 4 A would be more informative with a measurement of increasing laccase activity in culture supernatants.  It is not possible to judge if there really is more laccase produced after 72 h from the gel presented.
  5. Line 232.  Again the authors describe the enzyme as psychrophilic without evidence.  It is crucial that the activity of the recombinant enzyme at lower temperatures be given in the paper.
  6. Table 4.  State the concentration of dye used.
  7. Line 402.  This is debatable and requires statistical analysis (t-test).  The standard error/deviation makes it difficult to determine if there is any significant difference.
  8. Lines 465-487.  The authors describe other biochemical approaches to vanillin production but do not state how the KbLcc1 is (or could be) better.

Round 2

Reviewer 1 Report

The revised version of the article by Wisniewska et al entitled “Novel cold-adapted recombinant laccase KbLcc1 from Kabatiella bupleuri G3 IBMiP as a green catalyst in biotransformation” has been greatly improved, as requested. Authors have clearly answered all the points raised and the final version of the article undoubtedly meets the standards for publication in IJMS.

However, there are still few minor things that I would like to pinpoint.

  • Please revise the article as there are a lot of cases of different words without a space between them. For example in line 21, “pH 3.5,with” should be changed to “pH 3.5, with” or in line 22 “reached14.39%.” should be changed to “reached 14.39%.” Check lines 67, 77, 94, 118, 220, 227, 230, 312, 335, 336, 350, 354, 381, 429, 430, 449, 461, 489, 521, 522, 531, 548, 567, 571, 592, 654, 703, 717, and TableS1.
  • Line 517. Change “evaluated. [26].” to “evaluated [26].”
  • Supplementary Tables and Figures titles seem to be in different sizes.
  • Line 245. Add (Figure 5c) after “recorded at 30ºC.”
  • Line 250. Add (Figure 5d) after “for 120 min”.
  • Check Figures and Tables numbering throughout the text. In line 254 Figure 5c should be Figure 5d.
  • Maybe I missed something but still I don't understand Figures 5a and 5b with regard to pH stability and ABTS. How is it possible to have such a different pH stability with ABTS after 24h incubation at 4ºC? How can be explained the high enzyme activity at pH 8 for ABTS?.
